# A Comparative Life Cycle Assessment of Palm Kernel Shell in Ceramic Tile Production: Managerial Implications for Renewable Energy Usage

Handaya [1,*], Marimin [2], Dikky Indrawan [1] and Herri Susanto [3]

1   School of Business, IPB University, Bogor 16680, Indonesia
2   Faculty of Agricultural Engineering and Technology, IPB University, Bogor 16680, Indonesia
3   Faculty of Industrial Technology, Bandung Institute of Technology, Bandung 40132, Indonesia
*   Correspondence: ir.handaya@gmail.com; Tel.: +62-811-8828-795

**Abstract:** The palm oil industry is a promising biomass source, as the production generates wastes more than four times that of the main product. In 2020, for 45 MT of crude palm oil production in Indonesia, it was estimated that around 12 MT of palm kernel shell were generated or equivalent to 5.4 MTOE in net calorific value. This high calorific value of solid waste can be used by industries as a source of renewable energy, once it is proven to be technically, environmentally and economically feasible. In this comparative study, the life cycle assessment method was deployed to determine the environmental feasibility of palm kernel shell usage as an alternative renewable energy source to coal and natural gas in ceramic tile production through the application of combustion technology. The novelty of this study lies in a cradle-to-gate approach by comparing the carbon footprint of biomass from agriculture industrial waste with common fossil fuels as sources of energy for a highly energy-intensive industry. This research demonstrates that by evaluating the total life cycle of a fuel, the perspective on environmental impacts can be quite different when compared to looking solely at the end-use process. This study shows how the deployment of life cycle assessment would create managerial implications toward the decision making of fuel selection with carbon footprint considerations.

**Keywords:** affordable clean energy; ceramic tile industry; industrial fuel innovation; life cycle assessment; palm kernel shell





## 1. Introduction

What distinguishes renewable and nonrenewable resources is that renewable resources must have a significant rate of renewal on a relevant economic time scale for which planning and management activities need to be undertaken [1]. Energy production from biomass, such as derivatives from oil palm tree, has an advantage in terms of economic time scale since it is renewable within several years compared to fossil fuels that take thousands or millions of years for reproduction [2–4].

Furthermore, there are ongoing concerns regarding the implications of greenhouse gas (GHG) build-up in the atmosphere coming from the exploitation of nonrenewable fuels, which is not just creating deteriorating impacts from an environmental standpoint [5], but also an economical one in term of costs to neutralize and stabilize $CO_2$ levels [6] and social impacts in the form of human health challenges, including the risks from heat stroke, degenerative illnesses and vector-borne diseases [7]. The Kyoto Protocol of the United Nations Framework Convention on Climate Change (UNFCCC), the first formalized global commitment in reducing GHG emissions, includes the Clean Development Mechanism (CDM), which is relevant in promoting the development of biomass as renewable energy sources [8].

With respect to economic value, it was estimated that biomass will contribute about 8% of the global energy usage in 2050 or equivalent to 50 quadrillion BTU, which will already be a quarter of current petroleum-based fuels consumption [9]. Assuming the price of fuel oil at USD 15 per million BTU as per the current market price and 50% of biomass-based fuel will be utilized by the industrial sector, this will represent a value of USD 375 billion per annum or around 1.6% of global GDP in 2021, which signifies a huge potential of industrial economic value that warrants further exploration.

The palm oil industry, as the industry generating palm kernel shell (PKS), is the largest biomass source from the agricultural industry in Indonesia with a growing utilizable quantity along with the steady growth of the palm oil demand [10–12], which makes these biogenic materials promising renewable and sustainable alternatives to fossil fuels. Beyond conversion of crude palm oil (CPO) into biodiesel, wastes from palm oil processing, which are principally biomass with high quantity and variety, can also be utilized both as liquid and solid fuels [10,11]. Hambali and Rivai [10] estimated that there are around 429 tons of biomass generated for 100 tons of CPO produced, which means more than four times the amount of waste is generated with most convertible into energy, of which 27 tons of waste are PKS. For 45 MT of CPO produced in 2020 [13], there were around 12 MT of PKS available to be utilized as a source of energy, which was equivalent to 5.4 MTOE in terms of net caloric value [14].

PKS is a potential substitute for fossil fuels for heat generation at the industrial scale with lower impact on the environment. PKS, with its high lignocellulosic content, can be converted into energy by direct combustion or as feedstock for thermochemical conversion [4,15]. Currently, PKS is mainly used in a typical palm oil mill as solid fuel for the in-house boiler, where the generated steam is further utilized for electricity production and fresh fruit bunch disinfection [16]. For the purpose of heat generation, besides the direct combustion method, the densification of PKS to form briquettes can be used as a solid fuel [17].

Biomass can be utilized for industrial purposes after it is proven to be technically, environmentally and economically feasible [18]. This research is intended to find out whether PKS is feasible from an environmental point of view in comparison with frequently used fossil fuels, namely, coal and natural gas, in ceramic tile production. The industry has a highly energy-intensive production process with total energy consumption of 2258 kcal/kg product [19] and generates as much as 16.42 kg $CO_2$ per sqm of produced tile [20], or approximately 263 Mt of $CO_2$ from the global production of 16 billion sqm in 2020 [21].

The life cycle assessment (LCA) with cradle-to-grave approach is the chosen method to determine the environmental feasibility of PKS by calculating GHG emissions at each stage throughout its end-to-end supply chain and comparing the emissions from coal and natural gas with the case study of the spray drying process in ceramic tile production. The study results are expected to facilitate the decision-making process with environmental impact considerations within the context of industrial fuel selection, more specifically for highly energy-consuming industries such as ceramic tile.

By conducting a comparative LCA study, a conclusion can be drawn in order to verify whether the whole supply chain operation will generate net emissions gain or reduction. Such a study was exemplified by Bird et al. [22] in the case of selecting bioenergy systems, Hoppe et al. [23] in the use of natural gas for transportation and chemical production, Olofsson and Börjesson [24] in relation to the use of residual biomass as resource, Nduagu et al. [25] in the iron and steel industry and Kar et al. [26] in the GHG effects of alternative biomass and fossil energy sources for district heating. In comparison with this study, none of the preceding papers show a distinctive comparison in term of cradle-to-grave carbon footprint between the usage of agriculture industrial waste and fossil fuels for the application of energy generation in a downstream industry.

## 2. Materials and Methods

### 2.1. Palm Kernel Shell, Other Oil Palm Biomass, and Comparison Fuels

From various literature, it was found that there are qualitative similarities between PKS and other oil palm solid biomasses in term of thermophysical and chemical properties, which suggests all of these wastes as good sources of energy, either for self-generated heat in the palm oil mill or to be utilized by other industries with thermal conversion processes [4,14,16–18,27,28]. Oil palm biomass essentially consists of carbon (C), hydrogen (H), oxygen (O), nitrogen (N) and some traces of sulfur (S), which are identified by the ultimate analysis methods. The composition of these elemental components determines the calorific value of the substance [29,30]. Table 1 reveals the typical ultimate and proximate analyses of oil palm biomass in comparison with coal and natural gas.

**Table 1.** Ultimate and proximate analysis of oil palm biomass vs. fossil fuels (% *w/w*).

| Oil Palm Biomass | C | H | O | N | S | FC | VM | Ash | MC | Reference |
|---|---|---|---|---|---|---|---|---|---|---|
| Palm kernel shell | 50 | 5 | 45 | 0.1 | 0.2 | 22 | 76 | 2 | 11 | [14] |
| Empty fruit bunch | 41 | 6 | 43 | 0.2 | 1.0 | 19 | 73 | 8 | 70 | [31] |
| Mesocarp fiber | 46 | 9 | 50 | 0.4 | 0.0 | 28 | 71 | 1 | 5 | [32] |
| Fronds | 45 | 5 | 49 | 0.7 | 0.1 | 15 | 81 | 4 | 8 | [33,34] |
| Trunks | 62 | 9 | 51 | 1.3 | 0.1 | 4 | 77 | 19 | 76 | [27,28,35] |
| Coal | 66 | 5 | 19 | 1.0 | 0.8 | 49 | 28 | 23 | 3 | [36] |
| Natural gas | 73 | 24 | 0.4 | 0.1–1.5 | <5 ppm | | | | | [37] |

Note: ultimate and proximate analysis (except MC) as dry basis.

It is known that oil palm biomass fixed carbon is constructed of lignocellulosic as a basic macro structure with some variations in the composition. More commonly, a proximate analysis, as shown in Table 1, also reveals three parameters to characterize a solid biomass, namely, fixed carbon (FC), volatile matter (VM) and ash content (ash), which all together along with moisture content (MC) will determine its calorific value and combustion properties.

Figure 1 shows the calorific values of oil palm biomass, both in higher heating value (HHV) and lower heating value (LHV), where PKS is by far the highest among oil palm biomasses. This is due to lower oxygen, lower ash and higher volatile matter contents in PKS, as predicted by the models from Ghugare et al. [29]. Furthermore, with lower nitrogen and sulfur contents, PKS will generate lower NOx and SOx emission, which is preferable from the viewpoint of environmental feasibility [38]. These findings ascertain the superiority of PKS over other oil palm biomasses to be utilized as industrial fuel through the direct combustion process.

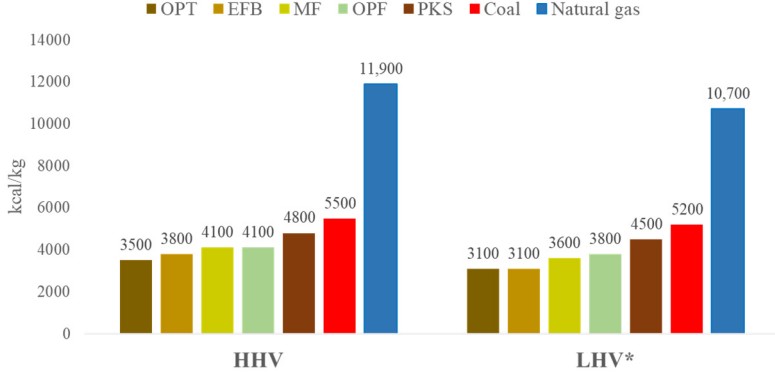

**Figure 1.** Calorific values of oil palm biomass and comparative fossil fuels. OPT [27]. EFB [31]; MF [32]; OPF [33]; PKS [14]; coal [36]; natural gas [37]. * LHV are calculated from LHV = HHV − (9 × %H + %MC) × 540.

Referring to the calorific values in Figure 1, PKS is the lowest among the alternative fuels, whereas natural gas yields the highest HHV with its highest level of carbon and hydrogen from the hydrocarbon content. With higher level of elemental carbon, lower oxygen and fixed carbon, it is predictable that coal provides higher calorific value compared to PKS. In general, coal also contains relatively lower moisture content than PKS, which leads to higher LHV. On the other hand, lower ash content in PKS will generate less solid waste post combustion compared to coal. One point to note is the different levels of volatile matter between PKS and coal, which needs to be taken into account in the design of combustion systems, particularly regarding the aspects of smoke and ignition control [39].

Given its unique physical, chemical and thermal characteristics, PKS delivers excellent combustion properties with relatively high calorific value and therefore it is a great potential fuel for heat and power generation [4]. With these properties, PKS has enormous potential to become a major source of energy for industrial usages. Despite PKS having lower calorific value compared to natural gas and coal, by investing in the right conversion technology, it can be utilized as an industrial fuel.

## 2.2. Life Cycle Assessment

In determining PKS environmental feasibility, the life cycle assessment (LCA) method was deployed in this research using data from various primary and secondary sources as listed in Table 2.

**Table 2.** LCA data source.

| Fuel | Process | Specificity | | | | | Type | Source |
|---|---|---|---|---|---|---|---|---|
| | | Very High | High | Medium | Low | Very Low | | |
| Palm kernel shell | Inbound land transport | | X | | | | Emission per km | [40] |
| | Sea transport | | X | | | | Emission per km | [41] |
| | Outbound land transport | | X | | | | Emission per km | [40] |
| | Combustion process | X | | | | | Concentration | field study |
| Coal | Land-use change | | | | X | | Complete unit | [42] |
| | Overburden processing | | | X | | | Concentration | [43] |
| | Coal processing | | | X | | | Concentration | [43] |
| | Inbound land transport | | X | | | | Emission per km | [40] |
| | Sea transport | | X | | | | Emission per km | [41] |
| | Outbound land transport | | X | | | | Emission per km | [40] |
| | Combustion process | X | | | | | Concentration | [19,20] |
| Natural gas | Land-use change | | | | X | | Complete unit | [42] |
| | Production and boosting | | | | X | | Concentration | [44] |
| | Processing | | | | X | | Concentration | [44] |
| | Transmission | | X | | | | Emission per km | [45] |
| | Combustion process | X | | | | | Concentration | field study |

The five phases of LCA in accordance with ISO 14040 standards [46], i.e., goal definition, scope definition, inventory analysis, impact assessment and interpretation, were conducted to analyze the carbon footprint of PKS in comparison with coal and natural gas from cradle to grave. The result of the interpretation phase is used as the basis of environmental performance evaluation for each alternative fuel. Figure 2 shows the relationship among these five phases.

With regards to the system boundary, the LCA studies for coal and natural gas commence with the generation stage, including the aspect of land-use change as a consequence of mining activities, whereas the starting point LCA for PKS needs to be defined differently. As PKS is a waste of oil palm processing, processes associated with PKS generation should not be accounted for in the assessment since it would have been produced regardless of the subsequent utilization [24,47,48]. Therefore, by this LCA accounting convention, the impacts on the carbon footprint related to oil palm plantation and palm oil milling are excluded in the LCA study of PKS.

According to ISO 14040, an LCA study can cater to various categories of environmental impacts, including resource use, human health and ecological consequences [46], which is commonly used as a decision-making support tool despite its limitations [49,50]. In this study, the impact is limited to the carbon footprint with GHG emissions as the output of

the study measured in $CO_2$ equivalent. This includes consideration that energy systems are the main source of GHG emissions responsible for climate change [48].

### 2.3. Spray Drying Process

As the case study, the spray drying of ceramic powder, illustrated in Figure 3 with process variables and parameters listed in Table 3, is the selected end-process unit equipped with the fuel combustion system as the source of hot gas for drying. In this process, the fuel is converted into heat via direct combustion and the generated hot gas dries the slurry that is fed to the spray dryer resulting in the ceramic powder for further processing to make ceramic tiles. The heat energy consumption in preparing powder through a spray dryer is typically 28% of the total heat energy used in producing ceramic tiles [20]. With a combination of high calorific value, lower ash and lower nitrogen and sulfur contents, PKS is a suitable solid fuel to be utilized in this thermal process through a proper selection of combustion technology [4].

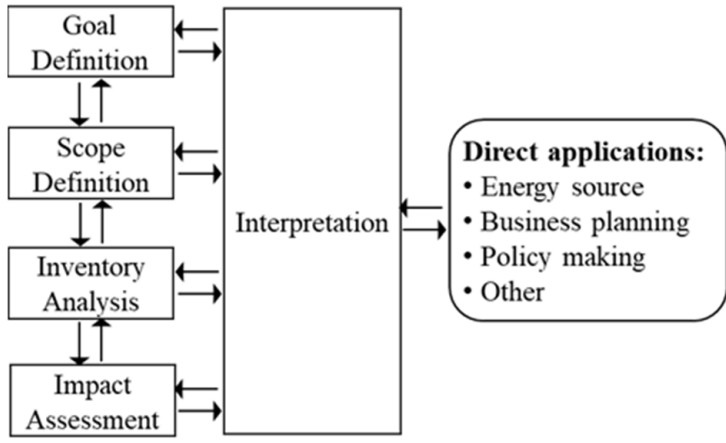

**Figure 2.** Life cycle assessment framework [46].

**Table 3.** Spray drying process variables and parameters.

| Variables/Parameters | PKS [1] | | Coal [2] | | Natural Gas [1] | |
|---|---|---|---|---|---|---|
| Basis of powder | 1000 | kg/h | 000 | kg/h | 1000 | kg/h |
| Powder moisture content | 7.3 | % | 7.0 | % | 8.0 | % |
| Slurry moisture content | 36.0 | % | 30.9 | % | 36.0 | % |
| Slurry temperature | 30 | °C | 30 | °C | 30 | °C |
| Fuel HHV | 4750 | kcal/kg | 5002 | kcal/kg | 9170 | kcal/Nm$^3$ |
| - moisture content | 22 | % | 8 | % | 0 | % |
| - hydrogen content | 9 | % | 5 | % | 24 | % |
| Fuel LHV | 4194 | kcal/kg | 4716 | kcal/kg | 8004 | kcal/Nm$^3$ |
| Fuel consumption | 91 | kg | 75 | kg | 45 | Nm$^3$ |
| Heat consumed | 278,657 | kcal/h | 210,984 | kcal/h | 272,475 | kcal/h |
| Net heat used | 3065 | kcal/kg | 2813 | kcal/kg | 6055 | kcal/Nm$^3$ |
| Thermal efficiency | 73 | % | 60 | % | 76 | % |

Sources: [1] Field observations; [2] [19,20].

The GHG emissions from combustion were calculated stoichiometrically based on the assumption of compositions listed in Table 1 and the measured thermal efficiencies from each fuel presented in Table 3, where the emission from PKS was compared with the emissions of coal and natural gas from the similar process of spray drying. The calculation was made with an assumption that the quantity of unburned carbon during the combustion process is negligible in comparison with the carbon emitted as $CO_2$ in the flue gas.

By definition, thermal efficiency is the ratio between the heat utilized (i.e., to evaporate water) and the energy supplied [51], which is calculated by the following formulae:

$$\eta_{Th} = \frac{Q_u/m_{fuel}}{LHV_{fuel}} \times 100\% \qquad (1)$$

$$Q_u = m_{H_2O,slurry}\, Cp_{H_2O}\left(100 - T_{slurry}\right) + m_{H_2O,evap}\, L_{vap,H_2O} \qquad (2)$$

$$LHV_{fuel} = HHV_{fuel} - \left[9\%H_{fuel}L_{vap,H_2O} + \%MC_{fuel}\left(Cp_{H_2O}\left(100 - T_{feed}\right) + L_{vap,H_2O}\right)\right] \qquad (3)$$

where:

$\eta_{Th}$ = thermal efficiency, %;
$Q_u$ = net heat used, kcal;
$m_{fuel}$ = fuel consumption, kg;
$LHV_{fuel}$ = lower heating value of fuel, kcal·kg$^{-1}$;
$HHV_{fuel}$ = higher heating value of fuel, kcal·kg$^{-1}$;
$m_{H_2O,slurry}$ = weight of water in feed, kg;
$m_{H_2O,evap}$ = weight of water evaporated, kg;
$Cp_{H_2O}$ = specific heat capacity of water, kcal·kg$^{-1}$·$^{\circ}$C$^{-1}$;
$L_{vap,H_2O}$ = latent evaporation heat of water, kcal·kg$^{-1}$;
$T_{slurry}$ = temperature of feed, $^{\circ}$C;
$\%H_{fuel}$ = fuel hydrogen content, %;
$\%MC_{fuel}$ = moisture content, %.

In measuring thermal efficiency for PKS and natural gas, data were taken from field observations during daily production operations for one whole month, each with the same spray dryer; whilst thermal efficiency for coal was calculated based on secondary data from a similar spray drying process of ceramic powder performed separately by Peng et al. [19] and Huang et al. [20].

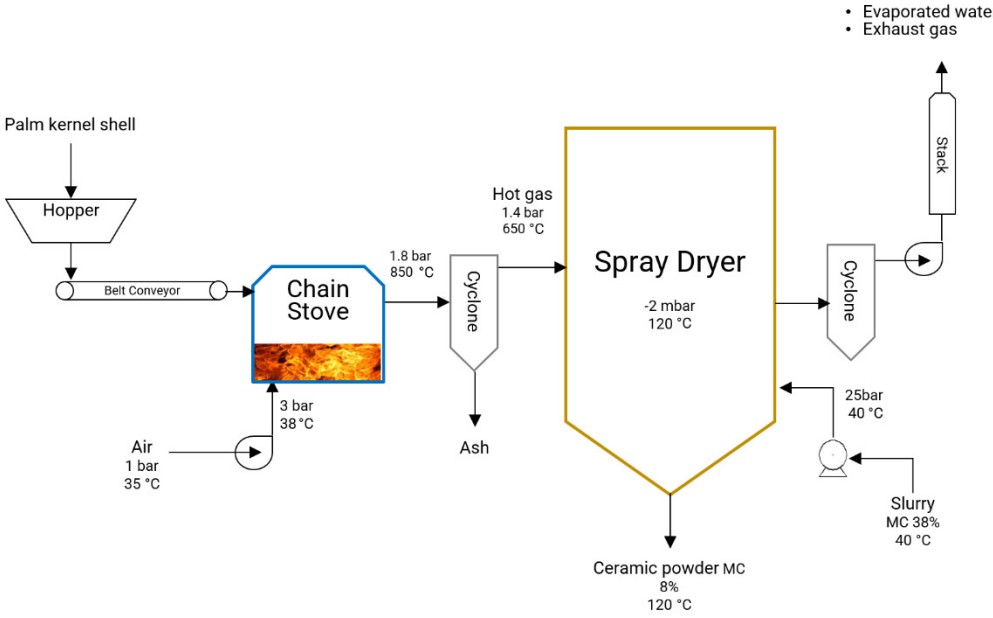

**Figure 3.** Spray drying process of ceramic powder.

## 3. Results

### 3.1. Goal Definition and Scope

The following set of goal definitions were identified for the LCA study in this research, namely:

- Intended application: to compare environmental impacts of industrial fuels among PKS and alternate fossil fuels, i.e., coal and natural gas, for application in industries with thermal processes by utilizing combustion technology.
- Limitation: the study only covers impacts to climate change referred to as carbon footprint, which are measured as $CO_2$ equivalent generation throughout the entire supply chain of the respective fuels.
- Decision context: supporting decision makers with recommendations for environmentally preferred fuel selection related to GHG emissions in industrial applications, whilst other aspects of environmental superiority and social impacts are out of context.
- Target audience: the study is intended for corporate decision makers to convince the usage of more environmentally friendly materials than fossil fuels and the academic community.
- Comparative studies: comparisons are made among life cycle assessment studies of alternative fuels from the combination of literature study and field observations.
- Commissioners: the study is self-funded with no conflict of interest and is commissioned by experts with the appropriate academic background.

The object of the LCA was PKS usage in the direct combustion process to generate heat for the purpose of ceramic powder spray drying in the case study, in comparison with alternative fossil fuels for the same heat load. The study deliverable is the carbon footprint from each fuel measured as $CO_2$ equivalent.

Prior to conducting the life cycle inventory analysis, a cradle-to-grave model, shown in Figure 4, was defined along with each fuel system boundary to define the scope of the LCA study and to capture all key processes within the supply chain of three alternative fuels. Data on material and energy inputs will be used later to calculate GHG emissions from each process.

As previously mentioned, while the PKS system boundary is limited only to distribution and usage, the system boundaries of coal and natural gas cover the whole chain process, including the land-use change as the consequence of mining setups.

In terms of geographical scope, Table 4 describes the selection areas of sourcing, the routes of distribution, and the location of end usage for each fuel.

**Table 4.** Life cycle inventory—geographical scope.

| Stage | Fuel | | |
|---|---|---|---|
| | **Palm Kernel Shell** | **Coal** | **Natural Gas** |
| Source | Riau Province, Sumatera | | Subang, West Java |
| Distribution | Inland Riau—Riau port to Jakarta port—inland Jakarta to Cikarang, West Java | | Piping from Subang to Cikarang |
| Use | Cikarang, West Java | | |

Cikarang, an industrial area located 40 km east of Jakarta, is the location of the industry where the case study took place. Riau province in Sumatera Island is selected as the largest producer of PKS, and as the source of coal in the consideration of comparative proximity with the source of PKS. Whilst Subang, a district in West Java, is selected as the nearest source of natural gas for industries in Cikarang. The route of distribution connecting the source points and the usage point currently exists.

*3.2. Life Sycle Inventory Analysis*

The life cycle inventory analysis commenced with the process identification in the fuel supply chains. This is done by elaborating the model and the system boundaries in Figure 4, by which calculations of GHG emissions can be performed based on material and energy balances in each process, in relation with the fuel requirement for heat generation in the ceramic tile industry.

As shown in Figure 5, for PKS, there are three unit processes related to transportations and combustion as the end process. In this model, the assumption is that GHG emissions

during PKS collection, loading/unloading and storing processes are negligible, since these processes only occur within limited distances and for relatively short periods of time.

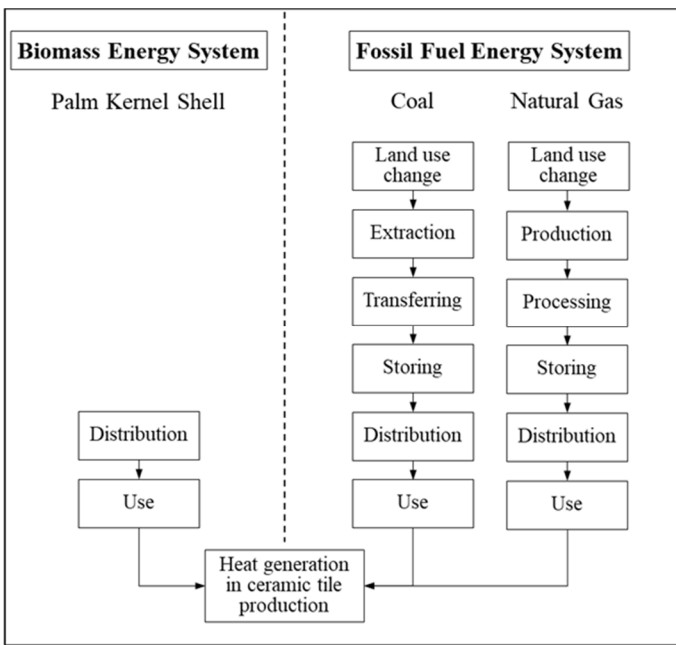

**Figure 4.** Life cycle inventory—system boundaries.

The transportation begins with inbound land transfer, where PKS stock is picked up from a palm oil mill and transferred by trucks to an inbound storage location usually near the inbound port. From there, PKS is then transferred to the port and loaded onto a barge. This whole inbound process takes place in the Riau province, Sumatera, with the total distance of 185 km. The over-water transfer takes place from the port of Riau to the port of Jakarta, crossing the Malacca Strait, South China Sea and Java Sea, reaching a distance of 712 nautical miles or 1319 km. The transportation process ends with the outbound land transfer from the port direct to the customer storage point or alternatively via an outbound storage in between the outbound port and the customer site, with the average distance of 58 km. The stored PKS is finally used for combustion at the customer site.

As for coal, the transportation and the end usage processes are assumed to be similar to PKS, whilst the mining operation includes overburden processing and coal processing in or near the coal mine. Another environmental impact that needs to be taken into account for coal comes from the land-use change.

The generation of natural gas commences with the production process to extract gas from a well. The gas is collected and then transferred to gathering and boosting networks, from which natural gas is further processed to remove impurities and then stored ready for distribution up to the customer site.

Unlike PKS and coal, natural gas undergoes a different route of transmission. From the main station, natural gas is transmitted through pipeline networks for distribution up to the end process unit. In this study, the pipe length for the transmission line was assumed to be 80 km, connecting the generation point at Subang and the consumption site at Cikarang, with a series of boosting stations along the network [45].

Furthermore, similar to coal, land-use change is another source of GHG emissions from natural gas extraction and production that needs to be accounted for. This is particularly important for the activities of onshore gas mining, such as the one in Subang, as the point of natural gas source in this study. Lastly, the GHG emissions for natural gas combustion are calculated by a similar method as for PKS and coal.

Based on the data sources in Table 2 and process identifications in Figure 5, life cycle inventory analyses were done for each process unit in the form of input-output diagrams,

where inputs are typical fuels for relevant process units (e.g., diesel oil for transport units) or the alternative fuel in the combustion process. Whilst the output is the quantity of alternative fuel from production and transportation units and the amount of heat generated from the combustion process. GHG emissions from each process unit are measured as $CO_2$ equivalent either per km for transportation units, per unit of alternative fuels mass for production units or per unit of energy for combustion processes. The results of the life cycle inventory analyses are shown in Figure 6 referring to each process unit involved in the utilization of PKS, coal and natural gas.

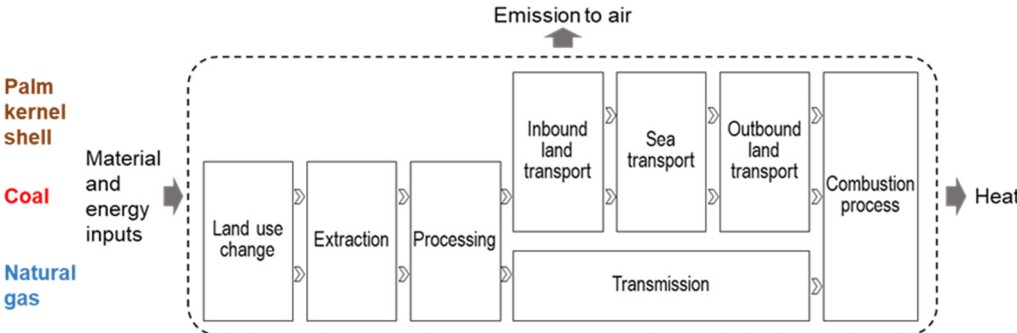

**Figure 5.** Life cycle inventory analysis—process identifications.

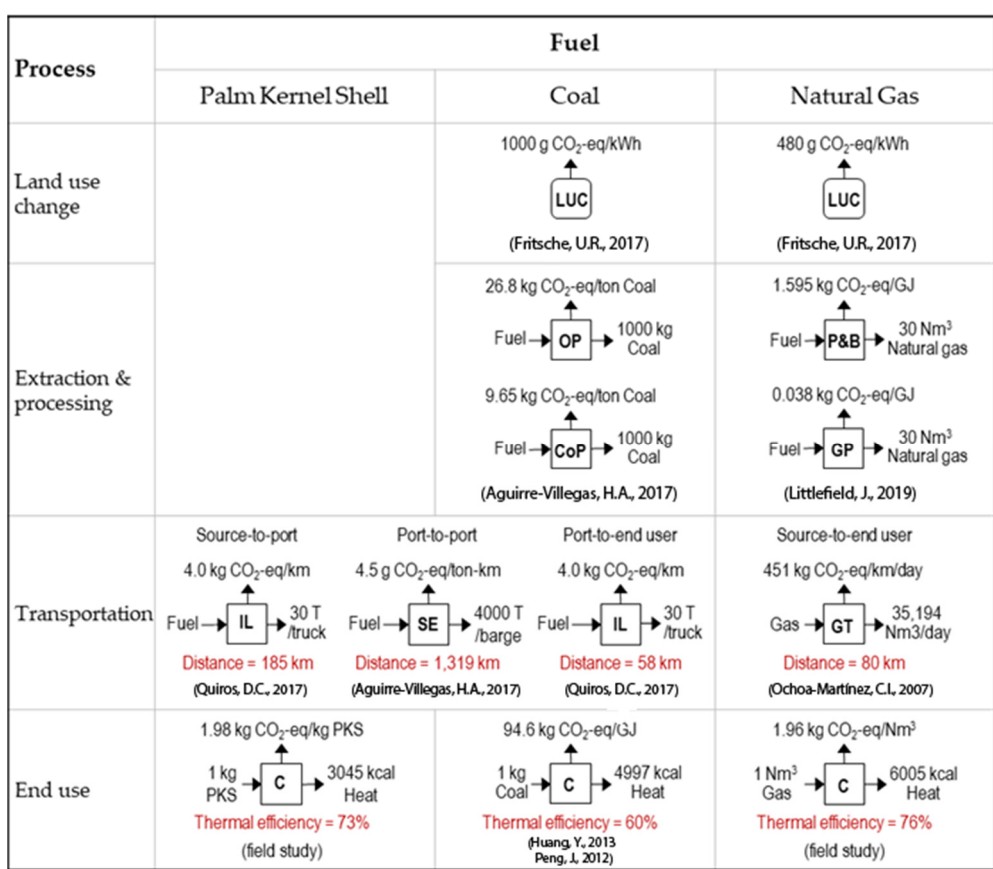

**Figure 6.** Life cycle inventory analysis results. In these flow diagrams, each box represents a process unit where LUC [42] denotes land-use change, OP: coal overburden processing, CoP: coal processing [43], P&B: gas production and boosting, GP: gas processing [44], IL: inland transportation [40], SE: sea transportation [41], GT: gas transmission [45] and C: combustion as the end process unit [19,20].

*3.3. Life Cycle Impact Assessment*

From the life cycle inventory analysis of each process unit, GHG emissions were calculated for each source of emission, i.e., land-use change and generation (both only for coal and natural gas), transferring and consumption processes, as summarized in Table 5. In conducting the impact assessment, all GHG emissions are calculated using the basis of net kWh heat generated by each alternative fuel from combustion.

**Table 5.** GHG emissions by sources in kg $CO_2$-eq/kWh heat.

| Activity | Fuel | | |
|---|---|---|---|
| | **Palm Kernel Shell** | **Coal** | **Natural Gas** |
| Land-use change | | 0.355 | 0.239 |
| Generation | | 0.005 | 0.001 |
| Transferring | 0.011 | 0.005 | 0.149 |
| Consumption | 0.566 | 0.605 | 0.284 |

As revealed in Table 5, for all three alternative fuels, the consumption processes are the highest sources of emissions, whilst land-use change is the second highest source in the case of coal and natural gas.

Further analyses of each alternative fuel resulted in the proportion figures of GHG emission sources shown in Figure 7. GHG emissions of PKS mostly come from the combustion process (98%) with only a small fraction (2%) from the transportation processes, where fossil fuel (namely, diesel oil) is consumed. For coal, despite the assumption of having the same travel distance, the impact of transportation is less than for PKS. This is due to higher calorific value per unit weight for coal in comparison with PKS, as depicted in Figure 1.

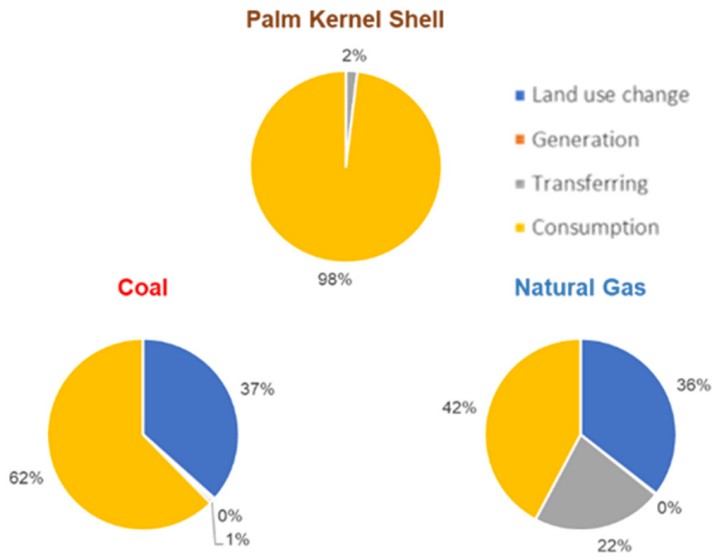

**Figure 7.** Proportion of GHG emission sources by fuel.

In terms of GHG emissions from transferring process, natural gas generates significantly higher emission compared to the other alternative fuels due to its pipeline transportation mode. Despite the advantages of being more continuous, stable and high-capacity, the pipeline transfer of natural gas poses a significant issue for GHG emission due to leakages that release methane mostly at compressor stations throughout the distribution network [45]. This factor diminishes the advantage of cleaner emission from combustion using natural gas.

By adding the emissions from all processes, the comparison of overall GHG emissions from the three alternative fuels are shown in Figure 8. From this cradle-to-grave assessment,

coal by far yields the highest level of GHG emissions followed by natural gas, whereas PKS generates the lowest emission in term of $CO_2$ equivalent per kWh heat making it the least impacting fuel to the environment in terms of carbon footprint.

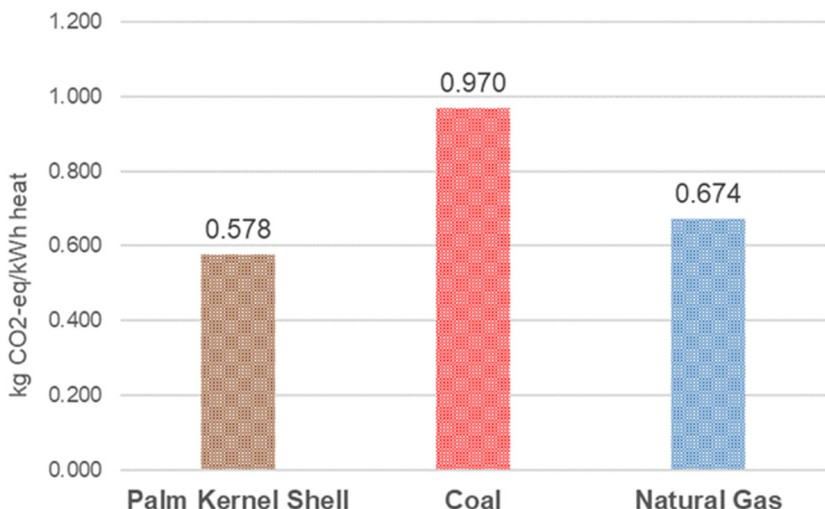

**Figure 8.** Fuel comparison—GHG emissions in kg $CO_2$-eq/kWh heat.

To calculate the overall GHG emissions in one calendar year from each alternative fuel, a basis of annual heat load needs to be determined. Based on the case study, such a basis was obtained from field measurement. The average daily evaporated water quantity is 332,684 kg $H_2O$ per day or equivalent with 204,600,473 kcal per day, after multiplying the quantity of evaporated water with the total sensible heat and evaporation heat of 615 kcal/kg $H_2O$. With the operation mode of 365 days in a year, the annual heat load is equal to 86,794 MWh. By deploying the LCA study results, annual GHG emissions from PKS, coal and natural gas can be calculated as depicted in Figure 9.

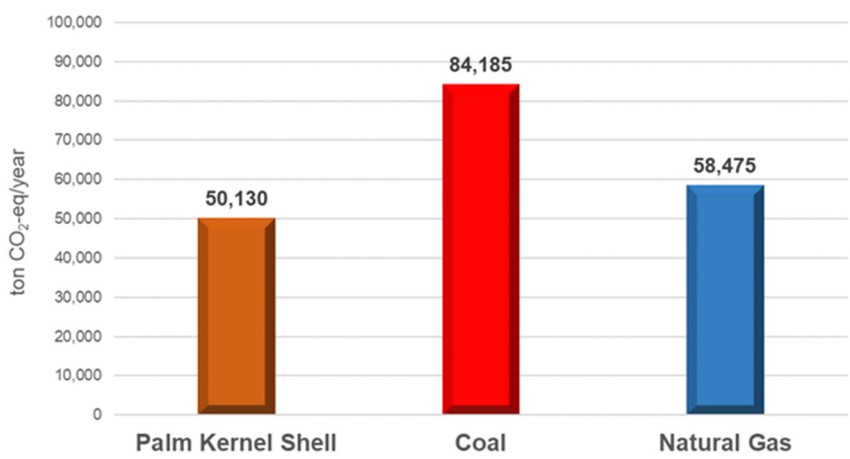

**Figure 9.** Comparison of yearly GHG emissions.

## 4. Discussion

This comparative study demonstrates that by evaluating the total life cycles, the interpretation of environmental impacts from each fuel can lead to different conclusions than by solely looking at the end process unit. GHG emissions from PKS and coal combustions are quite comparable, yet the impacts from land-use change, mining and collecting processes make coal a significantly less preferred fuel from the environmental viewpoint. Similarly, despite the process of natural gas combustion generating the least GHG emissions, the

whole process of getting the fuel into the end process unit yields a much higher carbon footprint in comparison with PKS.

The results from this LCA study validate the premise that PKS has superiority in terms of environmental impact over fossil fuels as the source of energy, particularly in the industrial context. Moreover, along with the steady growth of the palm oil industry, PKS will be a promising industrial fuel to replace coal and natural gas, at least partially, with recognizable advantages from its lower overall GHG emission, net zero carbon effect [52] and renewability due to the perpetual photosynthesis cycle in comparison to millions of years for fossil fuel formation [53]. By shifting the energy dependency of industries from fossil-based materials, the utilization of these materials can be directed towards more environmentally friendly usages, as suggested in the recent studies by Gagarin et al. [54] on coal for industrial materials and Hoppe et al. [23] on natural gas for chemical production.

Despite its demonstrated environmental feasibility, there is still an opportunity for improving PKS's combustion thermal efficiency, which will be beneficial not just because of lower $CO_2$ emission per unit heat but also from an economic aspect in terms of cost reduction. This can be achieved by optimizing the combustion process either through processing system improvements or utilizing more efficient technology, such as fluidized bed combustion [39]. Another way is to increase the energy density of the feed fuel by various pretreatment methods commonly used for biomass, such as briquetting [17,49] or fast pyrolysis [15]. Utilizing such improvements will further reduce the carbon footprint from higher heat utilization with similar levels of GHG emissions.

The lower greenhouse gas emission from PKS throughout its lifecycle is an appealing factor for industries to shift away from fossil fuels. The usage of PKS will promote business sustainability particularly in tackling the threatening issue of global warming. Given the relatively lower cost of PKS, as a waste from the palm oil industry, in comparison with the rising fossil fuel costs [4], this study has demonstrated that being more sustainable does not necessarily mean becoming less profitable. In fact, by pursuing initiatives towards the eradication of fossil fuel use, a firm can create even more value for its shareholders and society in general. In relation to the ceramic tile industry, this finding provides a viable solution to minimize the environmental impacts from high energy usage in its production while at the same time improving cost competitiveness.

The availability of raw materials in fostering the usage of biomass has been recognized by the Indonesian government as the main prerequisite for the investment of bioenergy [55]. Anwar et al. [56] pointed out that biomass from agroindustrial wastes with high lignocellulosic content, such as PKS, is a source of bioenergy, which is not just for meeting rising energy demand but will also help in avoiding and reducing environmental burden. This argument comes from the fact that these residual materials would have released GHGs into the atmosphere whether they were utilized as fuel or just disposed of in landfill. As evidenced from this research, the utilization of PKS as an industrial fuel will yield advantages in term of resource-use change (RUC) since it will lessen dependency on unsustainable fossil fuels and at the same time create environmental and economic benefits [24,57].

Nonetheless, the long-term availability of PKS as an industrial fuel is yet to be explored considering various inherent risks in its supply chain [4]. As pointed out by Dani and Wibawa [58], the existing policies and regulations on biomass energy in Indonesia have not yet been fully implemented, even though there is an abundance of biomass resources available to be utilized. Tasri and Susilawati [59] argued that a strong alignment, between renewable energy initiatives with government programs and support, is still required in order to optimize the use of biomass as an energy source in alignment with government commitments to carbon footprint reduction.

Furthermore, this study demonstrated the viability of deploying the LCA method in the assessment of a fuel carbon footprint in the context of managerial decision-making purposes. In spite of that, some potential biases need to be taken into account prior to expanding the method into a more complex decision-making situation, of which a trade-off

may need to be made between simplifying the procedure and augmenting the capabilities of LCA [50].

## 5. Conclusions

The interpretations from the LCA study in this research confirms that PKS has an environmental advantage in terms of GHG emissions, making it a feasible alternative fuel for industrial energy use, particularly in highly energy-intensive industries, such as ceramic tile manufacturing. This proven environmental feasibility will foster the shift towards renewable energy usage in the industrial sector. The shift will also be a sustainable solution to solving agriculture waste issues as the palm oil generating industry is continuously growing. This win-win situation will in turn accelerate the achievement of sustainability goals regarding net zero carbon emission.

This study shows that by evaluating the complete life cycle of a fuel, the perspective on environmental impacts can be quite different compared to only looking at the end-use process, which may affect the outcomes of business decision-making processes. From the environmental point of view, utilizing biomass from oil palm industry waste will give a net positive impact on the effort to reduce the carbon footprint. This research has proven that premise to some extent and this is even without considering the advantage of biomass carbon neutrality over fossil fuels.

Lastly, we would like to recommend further studies on how to increase thermal efficiency of PKS combustion and PKS supply continuity in the long run. By achieving improvements in these two areas, it will foster greater use of biomass in general and PKS in particular within highly energy-consuming industries, which in turn will accelerate the eradication of fossil fuel usage and save the environment.

**Author Contributions:** Conceptualization, H.; methodology, H.; formal analysis, H.; technical evaluation, H.S.; writing—original draft, H., writing—review and editing, D.I. and H.S.; supervision, M. All authors have read and agreed to the published version of the manuscript.

**Funding:** This research received no external funding.

**Institutional Review Board Statement:** Not applicable.

**Informed Consent Statement:** Not applicable.

**Data Availability Statement:** The data presented in this study are openly available in FigShare at https://doi.org/10.6084/m9.figshare.20278377.v1.

**Conflicts of Interest:** The authors declare that they have no known competing financial interests or personal relationships that could have appeared to influence the work reported in this paper.

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
