# Peer review of "A Comparative Life Cycle Assessment of Palm Kernel Shell in Ceramic Tile Production: Managerial Implications for Renewable Energy Usage"

_sustainability, doi:10.3390/su141610100_

Round 1
Reviewer 1 Report
The objective of this study is to evaluate the environmental impact of ceramic marble production. The study quantifies a number of environmental impact indicators as well as energy production from a lifecycle perspective. The article presents relevant results and findings supported by field studies that are worth publishing in this journal. However, important amendments must be made to improve the communication of the article.
Abstract - rather than too much of an introduction, can you showcase the research importance and research gap in the abstract. Then, how did you achieve the gap to showcase the results?
The Introduction section requires to be contextualized on a broader perspective, encompassing the environmental impacts of the Palm kernel shell production chain and use of ceramic marble tile from a global and/or regional perspective, it also requires to be more focused on the nature of the problem exposed as well as discuss the results of previous work concerning the impact of ceramic marble tile production elsewhere, specifically with regards to the carbon footprint and net carbon flux assessed in the article.
The research gap is not showcased well. You need to present this and that is what you can use to build your whole research work with aims, objectives, and so on. The authors need more international papers to support their introduction and results section.
Besides, it is desirable, given the extensive analytical work presented in the article, to include a Discussion section to expand the interpretation of the results, remark on the key findings, and state their implications from a broader perspective.
Conclusion and recommendations - This is more or less a summary. Can you present your sound conclusions? Not a place to discuss the results but a place to showcase your contribution.
References - Need recent and widely used references. Arrange them into the correct format too.
Reviewer 2 Report
This study conducted LCA of PKS. The following issues should be critical in this study.
(1) Allocation
PKS is a by-product of palm oil. By using allocation technique (e.g., economic allocation), the environmental burdens accounted for palm oil production should be allocated to PKS. According to Figure 4, palm oil production is not included in system boundary. Allocation is fundamental in the LCA. I think economic allocation technique should be appropriate in this case.
(2) Land use change
Inclusion of palm oil production in the system will bring the controversial issue in LCA; that is, inclusion of land use change (Please refer [Kosai, S. et al. Estimation of Greenhouse Gas Emissions of Petrol, Biodiesel and Battery Electric Vehicles in Malaysia Based on Life Cycle Approach, https://doi.org/10.3390/su14105783]). Since land use change is considered for coal and natural gas in the system, this also should be considered for PKS.
Reviewer 3 Report
This paper evaluates the total life cycle of PKS and finds that from the environmental point of view, utilizing biomass from oil palm industry waste will give a net positive impact in the effort to reduce carbon footprint.
The introduction section is good with preparation of detailed background. It includes required information to paper concept. Methodology is suitable to the topic. It was well-applied and well-presented during the paper. Furthermore, the method is supported with clear literature review and reasoning. LCA results are presented well by authors. Discussion is complete with managerial implication for industry as well as academic readers. Most of readers with the background of related issues may benefit from this paper through the way it explores.
Round 2
Reviewer 2 Report
Now it can be accepted.